# A geometric criterion for the optimal spreading of active polymers in porous media

Christina Kurzthaler [1,8 ✉], Suvendu Mandal [2,3,4,8 ✉], Tapomoy Bhattacharjee[5,6], Hartmut Löwen[2], Sujit S. Datta [7] & Howard A. Stone [1 ✉]

Efficient navigation through disordered, porous environments poses a major challenge for swimming microorganisms and future synthetic cargo-carriers. We perform Brownian dynamics simulations of active stiff polymers undergoing run-reverse dynamics, and so mimic bacterial swimming, in porous media. In accord with experiments of *Escherichia coli*, the polymer dynamics are characterized by trapping phases interrupted by directed hopping motion through the pores. Our findings show that the spreading of active agents in porous media can be optimized by tuning their run lengths, which we rationalize using a coarse-grained model. More significantly, we discover a geometric criterion for the optimal spreading, which emerges when their run lengths are comparable to the longest straight path available in the porous medium. Our criterion unifies results for porous media with disparate pore sizes and shapes and for run-and-tumble polymers. It thus provides a fundamental principle for optimal transport of active agents in densely-packed biological and environmental settings.

[1] Department of Mechanical and Aerospace Engineering, Princeton University, Princeton, NJ 08544, USA. [2] Institut für Theoretische Physik II: Weiche Materie, Heinrich-Heine-Universität Düsseldorf, 40225 Düsseldorf, Germany. [3] Physikalisches Institut, Albert-Ludwigs-Universität Freiburg, 79104 Freiburg, Germany. [4] Institut für Physik der kondensierten Materie, Technische Universität Darmstadt, 64289 Darmstadt, Germany. [5] The Andlinger Center for Energy and the Environment, Princeton University, Princeton, NJ 08544, USA. [6] National Centre for Biological Sciences, Tata Institute of Fundamental Research, Bangalore 560065, India. [7] Department of Chemical and Biological Engineering, Princeton University, Princeton, NJ 08544, USA. [8] These authors contributed equally: Christina Kurzthaler, Suvendu Mandal. ✉email: ck24@princeton.edu; suvendu.mandal@pkm.tu-darmstadt.de; hastone@princeton.edu

Microorganisms display agile motility features to optimize their survival strategies and efficiently navigate through their natural disordered and porous habitats[1–5]. While locomotion by swimming represents the most prominent, inevitable transport feature of many microorganisms, sudden changes of their swimming direction are also an essential tool for their efficient search for nutrients[6] or escape from harmful environments[7]. These reorientation events are generated by intrinsic biophysical mechanisms and generate different swimming modes, such as the run-and-tumble motion of *Escherichia coli*[3] or *Bacillus subtilis*[8], run-reverse(-flick) patterns of diverse bacteria[9,10], sharp turns in swimming algae[11], and run-reverse behavior of different species of archaea[12]. In unconfined media these transport features lead to trajectories reminiscent of a random walk, yet their consequences for the navigation through real, porous environments, characteristic of a wide variety of biological, biomedical, and environmental contexts, such as biological gels and tissues or environmental soils and sediments, remain largely unexplored. Understanding the underlying physical mechanisms is thus paramount for revealing fundamental microbiological processes, such as biofilm formation and community ecology[13,14], and has significant potential to enable novel nanotechnological applications[15,16].

Engineering the propulsive mechanisms of microorganisms has proven to be a promising route towards the design and development of smart, self-propelled cargo-carriers[17,18] that overcome several limitations of their passive counterparts (e.g., ordinary colloids). Yet, their ability to self-propel might not suffice to make them generally suitable for performing complex tasks in biomedical and environmental settings, where, for example, they may be expected to deliver drugs to a specific target[19], penetrate the porous structure of tumors[20,21], or find and induce degradation of contaminants[16,22]. In fact, randomizing the swimming direction of these autonomous agents could be an efficient strategy for reaching a target. To date, however, experimental realizations of controlled reorientation of self-propelled synthetic agents are sparse[23,24].

Experiments of biological microswimmers[25–28] and synthetic, active agents[29,30] in confined, disordered environments are often concerned with near-surface motility. These studies display a range of unusual phenomena, ranging from the circular swimming motion of bacteria near walls[25,26], hydrodynamic trapping[29,30], and enhancement of bacterial transport near surfaces due to the presence of obstacles[27]. Similarly, theoretical studies on active transport in crowded environments mainly focus on 2D models for active, point-like particles moving in periodic structures[31], on a lattice with obstacles[32], or disordered environments[33]. Accounting for elongated shapes, Brownian dynamics simulations of self-propelled flexible polymers have revealed subdiffusive motion in 2D porous media[34]. Quantitative studies of active transport in 3D porous media, however, are sparse and it was only recently that the hop and trap mechanism of individual *E. coli* cells moving in a 3D porous structure was identified[4,5].

While such studies shed light on how pore-scale confinement influences bacterial motility, it is still unclear what motility patterns are optimal for spreading in porous media. A clue comes from the seminal work of Wolfe and Berg[1], who studied the spreading of engineered bacteria, which lacked the ability to sense chemical gradients and whose tumbling rate could be controlled chemically. Their experiments indicated that smooth swimming strains of *E. coli* get stuck in the porous structure of the semi-solid agar, similarly to incessantly tumbling cells, while at an intermediate tumbling rate bacterial transport appeared more efficient. A non-monotonic transport behavior as a function of the tumbling rate has been found also theoretically in 2D systems[32,35,36], where the effects of pore shape[36] and mobile obstacles[32] have been addressed. However, the underlying optimal transport mechanism, dictated by the interplay of swimming characteristics and geometric features of the 3D porous medium, remains an open question.

In this work, we elucidate the spreading of self-propelled stiff polymers, as model systems for elongated microorganisms, in porous media by performing Brownian dynamics simulations and developing a coarse-grained theory. We demonstrate that reorientation mechanisms are indispensable for efficient dispersion through porous media, as intrinsic reversals enhance the overall diffusivities by up to two orders in magnitude. We identify a competition between the pore length, a direct measurement of straight pathways available in the pore space, and the run length of self-propelled polymers. In particular, the hopping lengths of rarely and frequently reversing polymers are constrained by the pore length and their intrinsic run length, respectively. Most importantly, maximal spreading occurs when the intrinsic run length of the polymers is comparable to the longest pore length of the porous medium, which allows us to introduce a simple but robust geometric criterion for optimal transport. Subsequently, we rationalize the non-monotonic transport behavior in terms of a renewal theory.

While such a non-monotonic behavior is predicted in refs. [32,35,36], our study unravels the underlying mechanism and demonstrates that this large-scale non-monotonic behavior persists irrespective of the pore shapes and reorientation mechanism of the polymers and is dictated by the maximal pore length only. These findings together with our geometric criterion provide fundamental, physical insight into earlier experimental observations[1] and thereby should guide the future design of synthetic cargo-carriers, applicable in biomedical and environmental settings.

## Results

**Model: run-reverse polymer in a porous environment.** We model the elongated shape of a bacterial cell by a stiff polymer with aspect ratio $L/\sigma = 5$, where $L$ denotes the polymer length and $\sigma$ the diameter of the individual monomers. We employ Brownian dynamics simulations of the discretized polymer, where the monomers are connected via stiff springs according to the well-established bead-spring model (see Fig. 1a and Methods). The stiffness of the polymer is characterized by a persistence length $\ell_p$, which exceeds its contour length $L$, $\ell_p/L \gg 1$. The self-propulsion of the polymer is modeled by active forces of magnitude $F_p$ acting on each monomer in the direction tangential to its backbone[37] (Fig. 1a). The velocity along the contour of the polymer is then determined by the friction coefficient $\zeta$ of a single monomer, $v_c = F_p/\zeta$, and each monomer is subject to translational diffusion with diffusivity $D_0 = k_B T/\zeta$, where $k_B$ is the Boltzmann constant and $T$ the temperature. The diffusive time scale of a single monomer is $\tau_0 = \sigma^2/D_0$.

In addition, the active polymer randomly reverses its swimming direction at exponentially distributed times with reversal rate $\lambda$ (Fig. 1b). At these 'pseudo' 'reversal events' the polymer either instantaneously changes its swimming direction and moves along the opposite direction or continues swimming along the same direction. The run length of the polymer is then defined as the length the polymer moves before it can reverse: $\ell_{\mathrm{run}} \equiv v_c/\lambda$. A run-reverse mechanism is employed by several microorganisms[9,12], yet for studying the large-scale spreading of active agents in a porous environment we anticipate that it can be used to model run-and-tumble bacteria[3,8] since often the only way to escape narrow pores is by reversing the swimming direction[4,5]. We demonstrate this by complementing our findings for run-reverse polymers by run-and-tumble polymers, which change their swimming direction randomly[3,8] (see Methods).

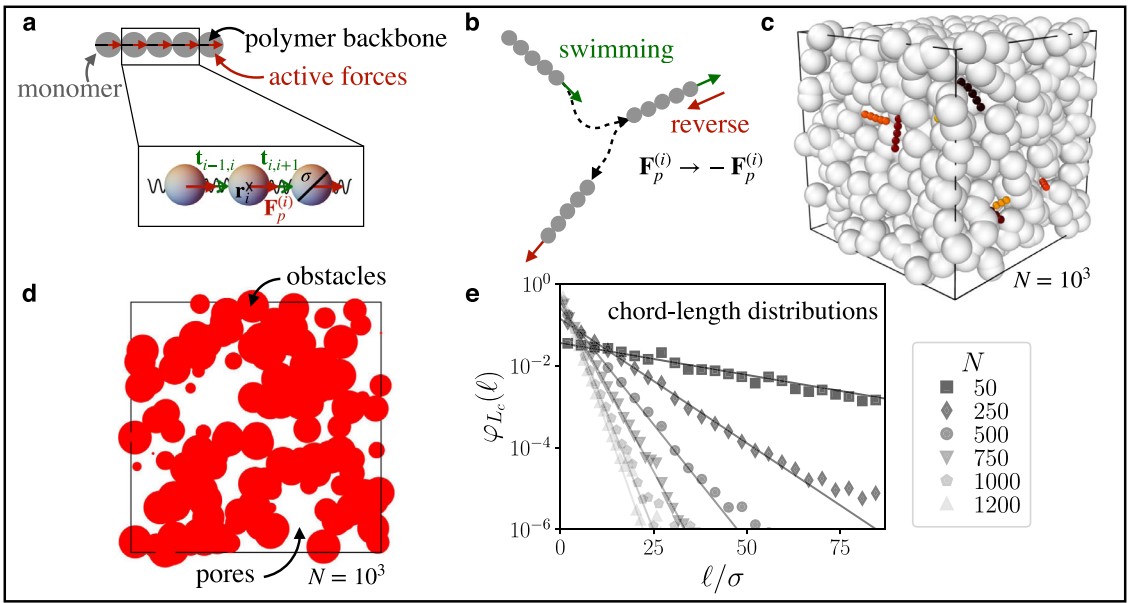

**Fig. 1 Model set-up of a run-reverse polymer in a porous environment. a** Sketch of a self-propelled stiff polymer composed of monomers (gray spheres) with active forces acting on each monomer in the direction tangential to the backbone of the polymer. (*Zoom*) The polymer chain is modeled using a bead-spring model, where $\mathbf{r}_i$ is the position of bead $i$ with diameter $\sigma$, and $\mathbf{t}_{i,i+1}$ is the tangent vector between bead $i$ and $i + 1$. The beads are connected by elastic springs. Furthermore, $\mathbf{F}_p^{(i)} = F_p(\mathbf{t}_{i-1,i} + \mathbf{t}_{i,i+1})$ denotes the active force acting on bead $i$. **b** Schematic of the run-reverse mechanism of the active polymer. At the run-reverse event the polymer randomly reverses its swimming direction, in particular, the active forces randomly change sign, $\mathbf{F}_p^{(i)} \to -\mathbf{F}_p^{(i)}$. **c** Simulation snap-shot of several active stiff polymers immersed in a porous environment composed of overlapping spheres. **d** Slice of the 3D porous environment. Red circles indicate 2D slices of the porous structure and white areas correspond to the open pore space. **e** Chord-length distributions $\varphi_{L_c}(\ell)$ for porous environments composed of a different number of $N$ overlapping spheres. Solid lines correspond to an exponential fit of the data. Source data are provided as a Source Data file.

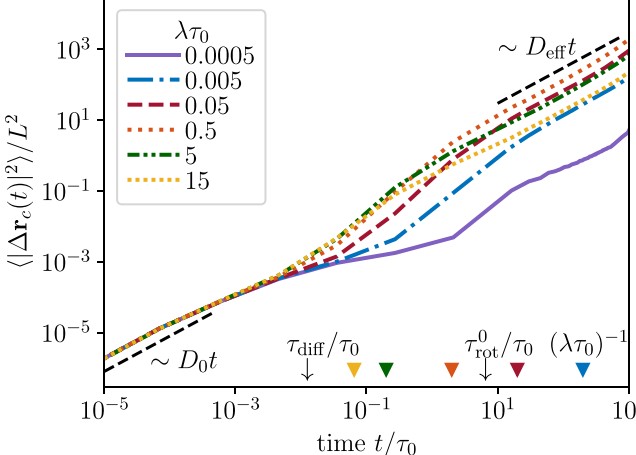

**Fig. 2 Mean-square displacements, $\left\langle |\Delta\mathbf{r}_c(t)|^2 \right\rangle$, of the center monomer of a polymer with different reversal rates $\lambda$.** Here, the polymer has 5 monomers and the Péclet number is Pe = 50. Further, $\tau_0$ denotes the diffusion time, $\tau_{\mathrm{rot}}^0$ the rotational relaxation time of a non-tumbling polymer, $\tau_{\mathrm{diff}}$ the cross-over time between short-time diffusion and directed swimming motion, and $L$ the polymer length. The colored triangles indicate the characteristic time of tumbling, $1/\lambda$ divided by $\tau_0$, for different reversal rates. Source data are provided as a Source Data file.

The swimming characteristics of run-reverse polymers can be described by two dimensionless parameters: (1) the Péclet number $\mathrm{Pe} \equiv v_c L / D_0$, which measures the self-propulsion strength relative to diffusion, and (2) the reversal rate $\lambda$ with respect to the characteristic diffusive time scale of a single monomer $\tau_0$, i.e., $\lambda\tau_0$.

To model the porous medium, we generate a disordered, monodisperse, porous structure composed of $N$ overlapping spheres of size $4\sigma$ within a cubic box of length $30\sigma$ [Figs. 1c and d for $N = 10^3$]. The micro-architecture of the porous medium can be characterized by the distribution of the straight paths, referred to as chord lengths, available in the medium, $\varphi_{L_c}(\ell)$[38]. It is shown for different $N$ in Fig. 1e. We further introduce the maximal chord length $L_{c,\mathrm{max}}$ via $\varphi_{L_c}(L_{c,\mathrm{max}}) = 10^{-5}/\sigma$. Throughout the manuscript we mainly focus on very densely packed environments and hence $N = 10^3$, unless stated otherwise. For this case, the maximal chord length is $L_{c,\mathrm{max}} \simeq 20\sigma$ and the medium is characterized by narrow channels of average pore diameter $\sigma_p$ comparable in size to the individual monomer $\sigma_p/\sigma \simeq 3.5$ (or the polymer $\sigma_p/L \simeq 0.7$), as is typical of many natural environments[39] (see Methods).

**Dynamics: mean-square displacement.** To investigate the dynamics of a self-propelled polymer in a porous environment, we measure the mean-square displacement (MSD) of the center monomer $\langle |\Delta\mathbf{r}_c(t)|^2 \rangle$ with $\Delta\mathbf{r}_c(t) = \mathbf{r}_c(t) - \mathbf{r}_c(0)$ and $\Delta\mathbf{r}_c(t) = [\Delta x_c(t), \Delta y_c(t), \Delta z_c(t)]^T$. We keep the Péclet number fixed, $\mathrm{Pe} = 50$, and tune the rate of reversal events, $\lambda$ (Fig. 2). At short times $t \lesssim \tau_{\mathrm{diff}} \equiv D_0/v_c^2$, the MSDs of active agents with different reversal rates $\lambda$ collapse and display a linear increase reflecting their diffusive motion at short times, which remains independent of the porous structure of the environment. At intermediate times, the directed motion of the polymers dominates and the MSDs exhibit a superdiffusive increase (for $\lambda\tau_0 \gtrsim 5 \cdot 10^{-4}$), which varies for different $\lambda$. The MSDs eventually cross over to a linear regime, which characterizes the effective diffusive behavior of the run-reverse polymers in a porous medium.

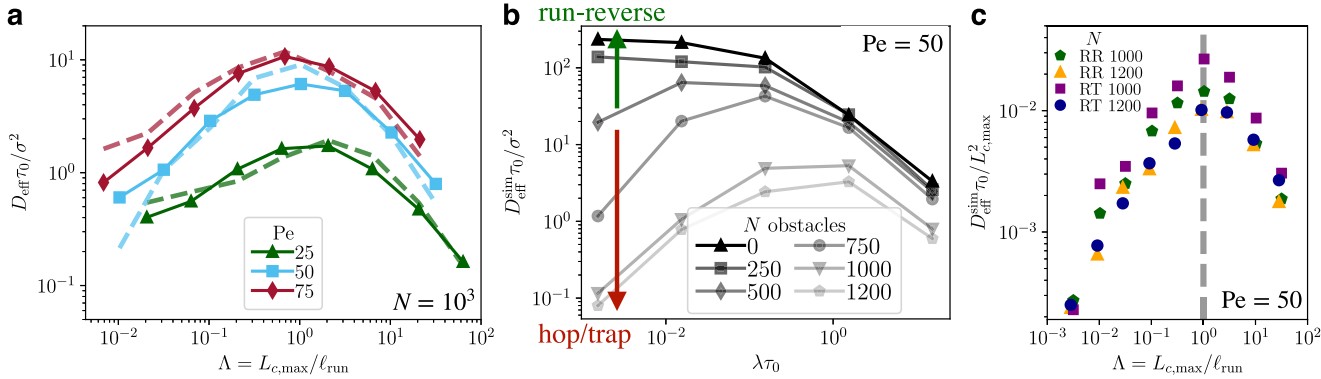

**Fig. 3 Effective diffusivities of active polymers spreading in porous media. a** Effective diffusivities, $D_{\mathrm{eff}}^{\mathrm{sim}}$, extracted from the simulations (filled symbols, solid lines) as a function of the scaled path length $\Lambda$ determined by the trade-off between the maximal chord length $L_{c,\mathrm{max}}$ and the run length of the polymer $\ell_{\mathrm{run}} = v_c/\lambda$, for different Péclet numbers. Dashed lines correspond to the theoretical predictions [equation (5)] of the hop-and-trap model with parameters extracted from individual trajectories. **b** Effective diffusivities, $D_{\mathrm{eff}}^{\mathrm{sim}}$, as a function of the reversal rate $\lambda\tau_0$ for porous environments with different number of obstacles $N$. **c** Rescaled effective diffusivities, $D_{\mathrm{eff}}^{\mathrm{sim}}$, of run-reverse (RR) and run-and-tumble (RT) polymers for Pe = 50 and different $N$. The data are rescaled by the maximal pore length $L_{c,\mathrm{max}}$ extracted from the chord-length distributions in Fig. 1e. Source data are provided as a Source Data file.

In contrast, non-reversing polymers can only reorient due to the interplay of thermal fluctuations, activity, and conformational chain dynamics[37]. The corresponding rotational relaxation time of the polymer is denoted by $\tau_{\mathrm{rot}}^0$ and can be extracted by measuring the fluctuation of the end-to-end direction of the polymer, $\mathbf{e}(t) = [\mathbf{r}_5(t) - \mathbf{r}_1(t)]/|\mathbf{r}_5(t) - \mathbf{r}_1(t)|$, which evolves as $\langle|\Delta\mathbf{e}(t)|^2\rangle = 2 - 2\exp(-t/\tau_{\mathrm{rot}}^0)$[40]. The MSD for rarely reversing polymers, $\lambda\tau_0 = 5\cdot10^{-4}$ (and polymers with $\lambda\tau_0 = 5\cdot10^{-3}$), displays a subdiffusive behavior at intermediate times, $t \sim \tau_{\mathrm{rot}}^0$, where the effect of the porous environment becomes apparent as the polymer slows down. At long times, the polymer has merely moved a distance comparable to its own length over the whole simulation time, which is a fingerprint of confined transport and indicates that motion in tight narrow spaces becomes hindered if active agents cannot reverse efficiently.

Moreover, we find that the cross-over time to long-time diffusion is determined by the faster of the two times: the reorientation time due to reversing $1/\lambda$ and the orientational relaxation time $\tau_{\mathrm{rot}}^0$ (indicated in Fig. 2). To quantify the long-time behavior, we extract the effective diffusivities via

$$D_{\mathrm{eff}}^{\mathrm{sim}} \equiv \lim_{t\to\infty} \frac{\langle|\Delta\mathbf{r}_c(t)|^2\rangle}{6t}. \qquad (1)$$

Our findings demonstrate that the long-time effective diffusivities can be enhanced by more than two orders of magnitude (over the whole simulation time of $10^3\tau_0$) upon introducing a reversal mechanism, $D_{\mathrm{eff}}^{\mathrm{sim}}(\lambda\tau_0 = 0.5)/D_{\mathrm{eff}}^{\mathrm{sim}}(\lambda\tau_0 = 0.0005) \simeq 4\cdot10^2$. This is in stark contrast to the motion of run-reverse polymers in a dilute environment, whose long time transport becomes suppressed upon increasing the reversal rate, $D_{\mathrm{eff}}^{\mathrm{sim}}(\lambda\tau_0 = 0.5)/D_{\mathrm{eff}}^{\mathrm{sim}}(\lambda\tau_0 = 0.0005) \simeq 3\cdot10^{-1}$.

**Non-monotonic transport behavior: effective diffusion.** Most prominently, the MSDs indicate that the long-time diffusivities $D_{\mathrm{eff}}^{\mathrm{sim}}$ of run-reverse polymers display a non-monotonic behavior with respect to the reversal rate $\lambda$. We further introduce the *scaled path length* $\Lambda = L_{c,\mathrm{max}}/\ell_{\mathrm{run}} = L_{c,\mathrm{max}}\lambda/v_c$, which characterizes the trade-off between the maximal pore length of the medium and the run length of the polymer. We find that the non-monotonic behavior persists for a broad range of Péclet numbers Pe = 25, … 75 (Fig. 3a filled symbols and solid lines). Most significantly, the effective diffusivities for all Péclet numbers display a prominent

maximum, where the scaled path length is

$$\Lambda = \frac{L_{c,\mathrm{max}}}{\ell_{\mathrm{run}}} = \mathcal{O}(1). \qquad (2)$$

Hence, we propose that equation (2) serves as geometric criterion for optimal transport of active agents in porous media, which occurs when the run length $\ell_{\mathrm{run}}$ is comparable to the maximal pore length $L_{c,\mathrm{max}}$ characteristic of the porous the environment.

Further, we found that the non-monotonic transport behavior persists for porous media with fewer obstacles, $N = 500, 750$, but becomes rather weak for dilute environments, $N \lesssim 500$, where it approaches the monotonic behavior of active polymers in an unconfined environment, see Fig. 3b. The chord-length distributions, $\varphi_{L_c}(\ell)$ (Fig. 1e), show a strong decay for densely packed environments.

Furthermore, we have addressed the effect of run-and-tumble motion on the overall spreading of active polymers, where, instead of reversing, the swimming direction after the tumbling event is random. Our results (Fig. 3c) demonstrate that our geometric criterion remains valid. We observe that the overall diffusivities increase for $N = 1000$ with respect to those of run-reverse polymers, as 3D tumbling allows polymers to spread further. Most importantly, in dense porous environments ($N = 1200$) the effective diffusivities of run-and-tumble polymers collapse with those of run-reverse polymers. This indicates that the overall spreading in dense porous environments is characterized by hop-and-trap dynamics, which are fully determined by the geometry, irrespective of the re-orientation mechanism. It further emphasizes that indeed the longest pore length of the environment is the characteristic length scale dictating the transport behavior of active polymers and furthermore confirms our geometric criterion.

In contrast to previous work[36], we have also found non-monotonic spreading of active agents in porous environments with concave pore shapes (Fig. 5 in Methods). This highlights that solely the interplay of length scales dictates this intricate behavior and not the pore shape.

Our findings offer an interpretation for the observations of the seminal experimental work by Wolfe and Berg[1], who used engineered strains of non-chemotactic *E. coli*. The tumbling rate and thus the run length of these mutants can be tuned by varying the concentration of an external inducer (isopropyl β-D-thiogalacto-side (IPTG)) in the medium. Monitoring the flagella bundles of cells tethered to a surface suggested an increase of the tumbling rate upon increasing the concentration of IPTG. Qualitatively, the experiments

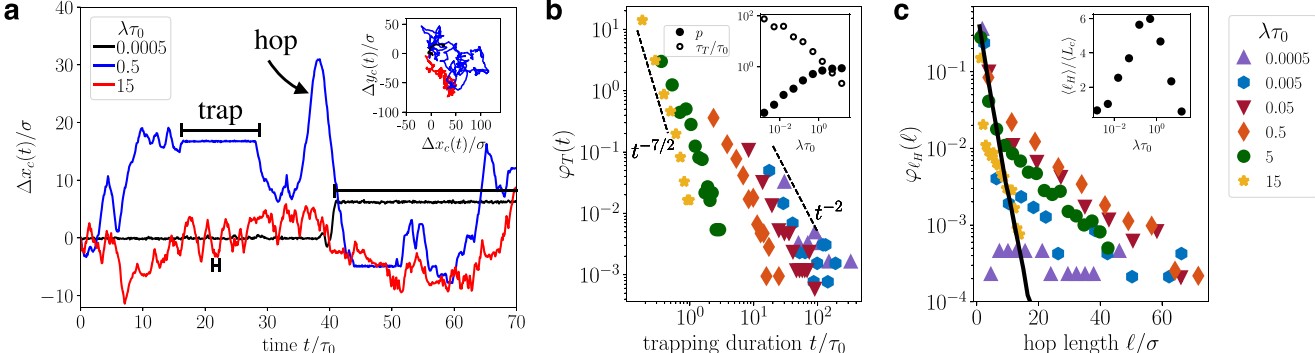

**Fig. 4 Polymer trajectories and distributions for the trapping time and hop length. a** Representative (1D) displacements, $\Delta x_c(t)$, of polymers with different reversal rates, $\lambda$, as a function of time for Pe = 50. Horizontal solid lines indicate the trapping phases for different $\lambda$. (Inset) Particle trajectories of the center monomer of a rarely reversing polymer, $\lambda\tau_0 = 5 \cdot 10^{-4}$ (black ; magnified by a factor of 4), and moderately reversing polymers, $\lambda\tau_0 = 0.5$ and 15 (blue and red, respectively). The trajectories are shown in the $xy$ plane, $(\Delta x_c(t), \Delta y_c(t))$. **b** Trapping time distribution $\varphi_T(t)$ and **c** hop length distribution $\varphi_{\ell_H}(\ell)$ for Pe = 50 and different reversal rates, $\lambda$, extracted from individual trajectories. The inset in **b** shows the fraction of time spent hopping $p = \tau_H/(\tau_H + \tau_T)$ and the average trapping time $\tau_T/\tau_0$ as a function of $\lambda\tau_0$. The black solid line in panel **c** indicates the chord length distribution of the medium $\varphi_{L_c}(\ell)$. The inset in **c** shows the average hopping length $\langle \ell_H \rangle$ normalized by the average chord length $\langle L_c \rangle$ of the porous medium. Source data are provided as a Source Data file.

of Wolfe and Berg revealed that bacterial swarms that tumble at an intermediate rate spread more than either incessantly tumbling cells or smooth swimming, i.e., non-tumbling, strains. However, the dynamical behavior of individual cells has not been quantified. Therefore, we anticipate that our simulations, which qualitatively confirm these experimental findings, enable us to elucidate the physics of the non-monotonic behavior and provide predictions for the optimal spreading.

**Individual trajectories**. To elucidate this non-monotonic behavior of the diffusivities, we investigate individual trajectories for Pe = 50 (Fig. 4a). For rarely reversing agents, $\lambda\tau_0 = 5 \cdot 10^{-4}$, the trajectories in the $xy$ plane indicate highly confined motion, whereas the trajectories of a polymer reversing at higher rates, $\lambda\tau_0 = 0.5$ and 15, cover significantly larger areas including many pores [Fig. 4a (inset)]. By inspecting the associated time evolution of the displacements, we observe that the motion of rarely reversing polymers $\lambda\tau_0 = 5 \cdot 10^{-4}$, i.e., the black curve in Fig. 4, is characterized by fast hopping events, at which the polymer moves through the pore space, and long, extended phases of trapping, where the polymer is trapped inside a pore. This motility pattern agrees with experimental findings of wild-type *E. coli* moving in a porous structure[4,5]. We further observe that upon increasing the reversal rate the trapping events become shorter, see blue curve for $\lambda\tau_0 = 0.5$ in Fig. 4a. In fact, they disappear for large reversal rates, $\lambda\tau_0 = 15$, where the trajectory is dominated by hopping events, i.e., the red curve in Fig. 4a.

Now, we explain the mechanism for the trapping and hopping, which identifies the main features necessary for an active agent to explore a porous medium. Suppose the active agent is exploring the porous environment with an intrinsic run time $\tau_{\text{run}} \sim 1/\lambda$ and after a while it enters a dead-end pore at a time $t$. Consequently, the agent will be trapped within the dead-end pore for the remaining time interval $[t, \tau_{\text{run}}]$. This implies that the trapping time of a self-propelled agent can potentially be reduced by increasing the reversal rate $\lambda$. The exemplary trajectory for $\lambda\tau_0 = 0.5$, which corresponds to the optimal reversal rate, exhibits the largest displacements with few trapping events (Fig. 4a). Moreover, we observe that despite the sparse and short trapping events of frequently reversing polymers, $\lambda\tau_0 = 15$, their hops become significantly shorter.

We quantify this behavior by extracting the distributions (1) $\varphi_T(t)$ for the trapping time, i.e., the time the polymer spends

trapped inside a pore, and (2) $\varphi_{\ell_H}(\ell)$ for the *hop length*, i.e., the distance the polymer moves from one trapping event to another or from one trapping event to the next reversal, from the individual trajectories (see Methods).

**Trapping time distribution**. We find that the trapping time distributions for active polymers with Pe = 50 exhibit a power-law scaling at long times with an exponent that increases from 2 to 7/2 for increasing reversal rates $\lambda$ (Fig. 4b). To rationalize this behavior, we extend the concept of an 'entropic trap' model introduced in the study of diffusing long polymers escaping small outlets[41] and recently for *E. coli* bacteria moving in a porous medium[4,5]. In this model, the escape of an active polymer from an obstruction in the porous medium is determined by the number of orientations ($\Omega_t$) keeping it trapped inside the pore and the number of orientations ($\Omega_e$) allowing the polymer to leave it. In short (see Methods), our model predicts a power-law trapping time distribution $\varphi_T(t) \sim t^{-(1+\beta)}$ at long times. Here, the exponent $\beta = X/C_0$ is related to the active energy $X$, which has dimensions of $F_p L$, and the average depth of the entropic trap $C_0$, which can be quantified by the average free energy difference between the two states: $C_0 = k_B T \langle \ln(\Omega_t/\Omega_e) \rangle$, where the brackets correspond to the ensemble average over all pores.

In accord with this phenomenological model, we find that the exponent $\beta$ increases for increasing reversal rate $\lambda$ (Fig. 4b). Specifically, at a fixed Pe (corresponding to $X = \text{const.}$), a larger reversal rate increases the probability $\Omega_e$ of leaving the (e.g., dead-end) pores and leads to a relatively lower trap depth $C_0$, and thus a larger exponent $\beta = X/C_0$. Further, we note that for $\beta \lesssim 1$, corresponding to the case where the active energy becomes smaller than the trap depth $X \leq C_0$, the entropic trap model predicts a divergence of the mean trapping duration $\tau_T \sim (\beta - 1)^{-1}$. This becomes evident in the monotonic increase of the mean trapping duration by orders of magnitude with decreasing reversal rates (Fig. 4b (inset) open circles).

**Hopping length distribution**. We further extract the distributions of the hopping length of the agents $\varphi_{\ell_H}(\ell)$ (Fig. 4c for Pe = 50) and rationalize that the hop lengths are determined by the interplay of the intrinsic run length of the active polymers $\ell_{\text{run}} \equiv v_c/\lambda$ and the pore geometry. In particular, we find that the chord length distribution (Fig. 4c black solid line) is approached

by the hopping length distribution of rarely reversing polymers, $\lambda\tau_0 = 5 \cdot 10^{-4}$ (and $5 \cdot 10^{-3}$). These agents have an intrinsic run length of $\ell_{run} \gtrsim 10^3\sigma$ (and $10^2\sigma$), which is much longer than the longest available straight pathways, $L_{c,max} \simeq 20\sigma$, and therefore their hopping motion is fully determined by the chord lengths of the porous medium (Fig. 4c). The distributions also show longer hops, however, the probabilities are rather low. Further, the average hop length is comparable to the average chord length for small $\lambda$, $\langle \ell_H \rangle \sim \langle L_c \rangle$ [Fig. 4c (inset)]. For large reversal rates $\lambda\tau_0 \gtrsim 5$ the distribution decays faster than that of polymers with an intermediate reversal rate $\lambda\tau_0 = 0.5$ and the average hop length approaches the run length of the polymer, $\langle \ell_H \rangle \rightarrow \ell_{run}$.

Our findings further show that polymers with an intermediate reversal rate, e.g., $\lambda\tau_0 = 0.5$, can follow the straight path available in the porous structure and continue to explore another successive pore without getting trapped, which leads to hopping lengths longer than the longest chord length. Furthermore, we find that the probability for longer hops becomes larger at an intermediate reversal rate than the chord-length distribution, which indicates that the polymer explores these more often than the shorter pores of the medium.

Our results also demonstrate that the probability for long hops and also the average hop length $\langle \ell_H \rangle$ [Fig. 4c (inset)] vary non-monotonically with $\lambda\tau_0$: they are lowest for $\lambda\tau_0 = 5 \cdot 10^{-4}$ and $\lambda\tau_0 = 15$ and highest for $\lambda\tau_0 = 0.5$. This strong non-monotonic variation of the average hop length could explain the optimal spreading and thus the maximal long-time effective diffusivity of run-reverse polymers in porous media (Fig. 3a). In particular, agents with Péclet number Pe = 50 and reversal rate $\lambda\tau_0 = 0.5$ have an intrinsic run length of $\ell_{run} = 20\sigma$, comparable to the longest pore length $L_{c,max}$. Our findings reveal that the active polymer continuously explores the pore space without getting trapped too often (Fig. 4a), which suggests the qualitative picture: the agent moves through the pores until it reaches an obstruction, where subsequent reversals allow it to continue to explore the pore space.

**Coarse-grained dynamics**. To quantitatively characterize the long-time effective diffusivities, we develop a coarse-grained model for the 3D dynamics of the alternating hopping and trapping phases. During the hopping phase the agent, modeled as a point particle, moves straight at effective velocity $v$ and reverses its swimming direction at rate $\lambda$ with propagator $\mathbb{P}_H(\Delta\mathbf{r}, t)$, which denotes the probability that the particle displaces $\Delta\mathbf{r}$ during time $t$ while hopping. During the trapping phase the particle cannot move and after the trapping event it hops along a new, random direction. The time the particle spends in a hopping or trapping phase is determined by the hopping and trapping time distributions extracted from the simulations (Fig. 4b, c). As the swim speed of the polymer during the hopping events remains roughly constant, the hopping time, i.e., the duration of a hopping phase of length $\ell$, $t \sim \ell/v$, also follows an exponential distribution $\varphi_H(t) = \exp(-t/\tau_H)/\tau_H$ with mean duration $\tau_H$. The trapping time distribution obeys a power-law behavior $\varphi_T(t) = \beta(1 + t/\tau)^{-1-\beta}/\tau$ with mean duration $\tau_T = \tau(\beta - 1)^{-1}$, where $\tau$ is a characteristic time scale for trapping. This power-law behavior suggests that the dynamics are non-Markovian and therefore we use a renewal theory[42,43].

The probability density $P(\Delta\mathbf{r}, t)$ for the particle to have displaced $\Delta\mathbf{r}$ during lag time $t$ is the sum of the probability densities for the particle to be in a hopping or a trapping phase: $P(\Delta\mathbf{r}, t) = P_H(\Delta\mathbf{r}, t) + P_T(\Delta\mathbf{r}, t)$. We further introduce the probability densities (per time) for the particle to start a hopping phase and a trapping phase at displacement $\Delta\mathbf{r}$ and at lag time $t$ by $H(\Delta\mathbf{r}, t)$ and $T(\Delta\mathbf{r}, t)$, respectively. Then the probability $P_T(\Delta\mathbf{r}, t)$ that the particle is trapped at $\Delta\mathbf{r}$ for lag time $t$ is obtained as the sum of the probability of never having hopped before $P_T^{(0)}(\Delta\mathbf{r}, t)$ and the probability of having hopped at least once before getting trapped at $\Delta\mathbf{r}$ and at an arbitrary earlier time $t - t'$:

$$P_T(\Delta\mathbf{r}, t) = P_T^{(0)}(\Delta\mathbf{r}, t) + \int_0^t dt' T(\Delta\mathbf{r}, t - t')\varphi_T^{(0)}(t'). \quad (3)$$

Here, $\varphi_T^{(0)}(t) = \int_t^\infty dt'\, \varphi_T(t') = [\tau/(\tau + t)]^\beta$ denotes the probability that the trapping time exceeds $t$. The probability to become trapped at $\Delta\mathbf{r}$ and time $t$ reads

$$\begin{aligned} T(\Delta\mathbf{r}, t) &= T^{(1)}(\Delta\mathbf{r}, t) \\ &+ \int_{\mathbb{R}^3} d\boldsymbol{\ell} \int_0^t dt' H(\Delta\mathbf{r} - \boldsymbol{\ell}, t - t')\mathbb{P}_H(\boldsymbol{\ell}, t')\varphi_H(t'), \end{aligned} \quad (4)$$

where $T^{(1)}(\Delta\mathbf{r}, t)$ denotes the probability for the first trapping event. The second term corresponds to the sum over the probabilities that the agent has escaped a trap and started to hop at $\Delta\mathbf{r} - \boldsymbol{\ell}$ at an earlier time, $t - t'$, until it gets trapped again at $\Delta\mathbf{r}$ and $t$. Similar equations hold for $P_H(\Delta\mathbf{r}, t)$ and $H(\Delta\mathbf{r}, t)$ (see equations (11)–(12) in Methods). We can derive an analytic solution for the probability density in Fourier–Laplace space, $P(k, s) = \int_0^\infty dt\, e^{-st} \int_{\mathbb{R}^3} d\Delta\mathbf{r}\, e^{-i\mathbf{k}\cdot\Delta\mathbf{r}} P(\Delta\mathbf{r}, t) \equiv \int_0^\infty dt\, e^{-st} \langle e^{-i\mathbf{k}\cdot\Delta\mathbf{r}} \rangle$[42]. By isotropy, it can be expanded for small wave numbers $k$ up to $\mathcal{O}(k^4)$ via $P(k, s) \simeq \int_0^\infty dt\, e^{-st}[1 - k^2\langle|\Delta\mathbf{r}(t)|^2\rangle/3!] = s^{-1} - k^2\langle|\Delta\mathbf{r}(s)|^2\rangle/3!$, which allows for an analytical derivation of the Laplace transform of the mean-square displacement $\langle|\Delta\mathbf{r}(s)|^2\rangle$.

Finally, we obtain the long-time transport behavior by taking the limit of $\langle|\Delta\mathbf{r}(s)|^2\rangle$ for $s \rightarrow 0$. We find $\lim_{s\to 0}\langle|\Delta\mathbf{r}(s)|^2\rangle \simeq 6D_{eff}^{theo}s^{-2}$, which corresponds to a long-time diffusive behavior $\langle|\Delta\mathbf{r}(t)|^2\rangle \simeq 6D_{eff}^{theo} t$ for $t \rightarrow \infty$ with effective diffusivity:

$$D_{eff}^{theo} = \frac{v^2\tau_H^2}{3(\tau_H + \tau_T)[1 + (1 - \cos\vartheta_0)\lambda\tau_H]}. \quad (5)$$

It is determined by the fraction of time spent hopping between traps, $\tau_H/(\tau_H + \tau_T)$, the hopping time, $\tau_H$, the effective velocity, $v$, and the tumbling rate $\lambda$. The tumbling angle for run-reverse motion, as employed here, is[9] $\vartheta_0 = -\pi$. The expression [equation (5)] depends on the exponent $\beta$ of the trapping time distribution via the average trapping time $\tau_T$. We note that for a power-law distribution truncated at rate $\gamma$, $\varphi_T(t) \sim \exp(-\gamma t)(1 + t/\gamma)^{-1-\beta}$, which may be more appropriate to describe our data (Fig. 4b), the effective diffusivity assumes the same form as in equation (5) with average trapping time $\tau_T \equiv \tau_T(\gamma, \tau, \beta)$ depending on the parameters $\gamma$, $\tau$, and $\beta$ (see Methods).

To compare the simulation data with our coarse-grained theory, we extract the average hopping and trapping times, $\tau_H$ and $\tau_T$, from the individual trajectories and obtain the effective diffusivity from the hop-and-trap model [equation (5)]. We observe that our model captures the non-monotonic trend of the simulation results, $D_{eff}^{sim} \sim D_{eff}^{theo}$ (Fig. 3a dashed lines). It is interesting that the hopping and trapping times, used as inputs for the model, can semi-quantitatively predict the non-monotonic trend of the effective diffusivities of a complex system. The deviations at small reversal rates, corresponding to the scaled path length $\Lambda \lesssim 10^{-1}$, are expected because at such small reversal rates the polymers remain trapped most of the time. Therefore, the extracted mean trapping times may be underestimated and even longer simulations would be required.

We rationalize the non-monotonic behavior by inspecting the fraction of time spent hopping, $p = \tau_H/(\tau_H + \tau_T)$ (Fig. 4b inset). As expected, for rare reversal events the fraction of time spent hopping is small $p \ll 1/2$. This corresponds to $\tau_H \ll \tau_T$ and therefore equation (5) reduces to $D_{eff}^{theo} = v^2\tau_H^2/(3\tau_T) \rightarrow 0$, as

large trapping times suppress transport. In contrast, the trapping times for frequently reversing polymers are negligible compared to the hopping times, $\tau_T \ll \tau_H$ as $p \gg 1/2$. Therefore, the behavior of the effective diffusivity is dominated by the reversal rate and equation (5) simplifies to $D_{\text{eff}}^{\text{theo}} \sim v^2/\lambda \to 0$, which vanishes as the reversal rate increases. Hence, maximal transport occurs at an intermediate reversal rate $\lambda$, in agreement with our simulation results for the long-time effective diffusivities and the average hopping lengths.

Our results indicate that the non-monotonic behavior of the effective diffusivities is a clear signature of hop-and-trap dynamics. This behavior vanishes for dilute environments, where the dynamics are solely determined by the run-reverse motion of the polymers. The transition between both motility modes is shown in Fig. 3b.

## Discussion

To rationalize the seminal experiments of Wolfe and Berg[1], which characterized spreading in a porous environment of bacteria with different tumbling rates, we have performed Brownian dynamics simulations of run-reverse stiff polymers in a porous environment and find a non-monotonic transport behavior as a function of the reversal rate. We introduce, for the first time, a geometric criterion for the optimal spreading of active agents in a porous environment, which occurs when the intrinsic run length is comparable to the longest straight path available in the environment. In particular, we demonstrate that this criterion remains valid in dense environments for different reorientation mechanisms of the active polymers. We further show that individual polymer trajectories exhibit a hop-and-trap mechanism, in accord with recent experiments of E. coli[4,5]. Our results suggest that optimal transport at an intermediate reversal rate is characterized by a maximal average hop length. In contrast, the motion of rarely and frequently reversing polymers is set by the pore length or the run-length, respectively. We corroborate these findings using a renewal theory for the coarse-grained hop-and-trap dynamics.

Our results demonstrate that this non-monotonic transport behavior of active agents in a porous environment persists irrespective of the details of the re-orientation mechanism and the shape of the pores. These findings indicate that the 'size of the pores, not their shape, matters' for this large-scale spreading and we therefore anticipate that this behavior is universal for densely packed environments. Nevertheless, this non-monotonic behavior fades for low packing fraction, which corroborates earlier predictions for dilute 2D porous environments with concave pore shapes[36].

In the future, it will be interesting to study the effect of swimmer shape anisotropy on active transport in a porous environment. In particular, recent work has shown that the non-monotonic behavior persists even for point particles on a 2D lattice with obstacles[32]. However, the interplay between pore geometry and run length remains to be explored. Most importantly, our study and recent experiments[4,5] predict power-law distributions for the trapping times, which could be amplified by particle shape. Taking into account this memory dependence in a coarse-grained description goes beyond earlier theoretical predictions[32,35].

We anticipate that our theoretical findings can be tested in various biological systems, such as bacteria[3,8–10], algae[11], or archaea[12]. This could shed light on fundamental microbiological processes, which include the adaption of the tumbling behavior under varying environmental conditions. Experimental observations of Bacillus subtilis[44] have shown that these cells can reverse their swimming direction once they encounter an obstacle and,

similarly, Spirochetes[45] display an increase of reversal rate while invading a heterogeneous fibrous medium. It would be interesting to elucidate if these spatially varying reversal mechanisms allow them to optimize their motion in a 3D porous medium. Similarly, our results could guide the design of future synthetic swimmers with, e.g., specific magnetic properties[46]. Their spreading could be enhanced by reorienting them via externally applied magnetic fields. On the macroscale, our findings could find application in instructing robots during search and rescue operations in disaster zones[36].

Beyond porous media, our results lay the foundation for studying transport of active semiflexible polymers in dynamically re-arranging environments composed of other polymers, such as the interior of cells[40,47]. The strongly interacting, crowded environment could lead to a relaxation of the extended trapping events of the active agents and entail complex entanglement effects of the individual constituents.

## Methods

**Brownian dynamics simulations of a stiff, run-reverse polymer in a porous environment**. We model the active polymer chain in terms of the well-known bead-spring model in 3D[37]. In particular, the polymer is a chain of contour length $L$ composed of $N_p$ spherical monomers with diameter $\sigma$ that are connected by springs (Fig. 1a). The monomers have positions $\mathbf{r}_i$ and the distance between two monomers is denoted by $r_{i,j} = |\mathbf{r}_j - \mathbf{r}_i|$, where $i,j = 1, \dots N_p$. We further introduce the tangent vector between two monomers by $\mathbf{t}_{i,i+1} = (\mathbf{r}_{i+1} - \mathbf{r}_i)/r_{i,i+1}$. The dynamics of the semiflexible polymer are described by the equations of motion for each monomer,

$$\zeta \frac{d\mathbf{r}_i}{dt} = -\nabla_i U + \mathbf{F}_p^{(i)} + \mathbf{F}_r^{(i)} \quad \text{for } i = 1, \dots N_p, \tag{6}$$

with friction coefficient $\zeta$, active forces $\mathbf{F}_p^{(i)} = F_p(\mathbf{t}_{i-1,i} + \mathbf{t}_{i,i+1})$, and stochastic forces $\mathbf{F}_r^{(i)}$ characterized by zero mean $\langle \mathbf{F}_r^{(i)} \rangle = \mathbf{0}$ and variance $\langle F_{r,j}^{(i)}(t) F_{r,k}^{(i)}(t') \rangle = 2k_B T \zeta \delta_{jk} \delta(t - t')$. The interaction energy, $U$, is characterized by the interaction between neighboring beads and the interaction between non-neighboring beads of the polymer chain itself and with the porous environment (see later section on details about the porous environment).

The elasticity of the chain is characterized by the well-established wormlike chain (WLC) model pioneered by Kratky and Porod[48]. The discretized interaction energy is

$$\frac{U^{\text{WLC}}}{k_B T} = -\frac{\ell_p N}{L} \sum_{i=1}^{N_p-2} \left( 1 - \mathbf{t}_{i,i+1} \cdot \mathbf{t}_{i+1,i+2} \right), \tag{7}$$

where we have introduced the persistence length $\ell_p$ of a semiflexible polymer. It is a measure of the decay length of its tangent-tangent correlations and allows distinguishing between flexible, $\ell_p/L \lesssim 1$, semiflexible, $\ell_p/L \simeq 1$, and stiff polymers, $\ell_p/L \gg 1$.

Interactions between neighboring beads are modeled by the finitely extensible non-linear elastic (FENE) potential,

$$\frac{U^{\text{FENE}}}{k_B T} = -\epsilon_{\text{FENE}} \sum_{i=1}^{N_p-1} \ln \left[ 1 - \left( \frac{r_{i,i+1}}{\delta} - \frac{L}{N_p \delta} \right)^2 \right], \tag{8}$$

which ensures a finite, maximal distance between two monomers, $L/N_p + \delta$ with $\delta = \sigma/4$. For monomer–monomer distances larger than $L/N_p + \delta$ the interaction potential diverges, $U^{\text{FENE}} = \infty$. Consequently, the entire polymer chain has a maximal length of $L + N_p \delta$. The interactions between non-neighboring polymer beads and the interactions between the polymer and the surrounding, porous environment are modeled using the Weeks–Chandler–Anderson (WCA) potential[49],

$$\frac{U_{ij}^{\text{WCA}}}{k_B T} = \epsilon_{\text{WCA}} \begin{cases} 4\left[ \left( \frac{d}{r_{ij}} \right)^{12} - \left( \frac{d}{r_{ij}} \right)^6 \right] & r < d \\ 0, & \text{else} . \end{cases} \tag{9}$$

Here, $d = \sigma$ for the interaction of two monomers and $d = (\sigma + \sigma_s)/2$ for the interaction between a monomer and an obstacle of the porous environment with diameter $\sigma_s$ (see later section on the *Porous environment*). The full contribution is the sum over all monomers and beads of the porous environment $U^{\text{WCA}} = \sum_{i,j;i\neq j} U_{ij}^{\text{WCA}}$.

The total interaction energy is then $U = U^{\text{WLC}} + U^{\text{FENE}} + U^{\text{WCA}}$, which includes (amongst others) three important dimensionless parameters: the persistence length of the polymer chain relative to the contour length $\ell_p/L$, and $\epsilon_{\text{FENE}}$ and $\epsilon_{\text{WCA}}$, which measure the relative importance of the interaction energies with respect to thermal energy. In the present study, we simulate stiff polymers with $\ell_p/L = 50$. To assure minimal polymer extension and prevent overlap of

**a**

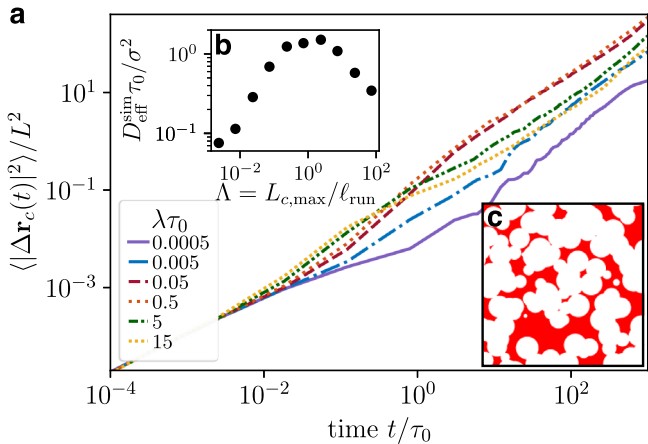

**Fig. 5 Active polymers in a porous environment with concave pore shapes. a** Mean-square displacements $\langle|\Delta\mathbf{r}_c(t)|^2\rangle$ of active polymers as a function of time for different reversal rates, $\lambda\tau_0$, and Péclet number Pe = 50. **b** Effective diffusivities $D_{eff}^{sim}$ as a function of $\Lambda = L_{c,max}/\ell_{run}$. **c** Slice of the 3D porous environment. Red areas indicate 2D slices of the porous structure and white areas correspond to the open pore space. Source data are provided as a Source Data file.

monomers and the environment, we set $\epsilon_{FENE} = 10^3$ and $\epsilon_{WCA} = 5$, respectively. Furthermore, we use a Brownian time step of $\tau_B = 10^{-6}\tau_0$ to integrate equation (6) using a modified version of LAMMPS[50] with $\tau_0$ the diffusive time scale of a monomer and wait for $10^9\tau_B$ before taking measurements.

In addition to self-propulsion, the polymer performs a pseudo run-reverse motion, as sketched in Fig. 1b. The reversal events occur at randomly drawn times, $t$, which follow an exponential distribution $\lambda\exp(-\lambda t)$ with reversal rate $\lambda$. At the run-reverse event, the active forces, acting along the polymer chain, randomly change sign, $\mathbf{F}_p^{(i)} \rightarrow \alpha\mathbf{F}_p^{(i)}$ with $\alpha$ randomly chosen from $\{-1, 1\}$. In our framework the reversal events occur instantaneously, so that the polymer does not stop before moving into the new direction.

### Porous environment.
The porous structure of the environment is generated by randomly distributed obstacles of diameter $\sigma_s$, which are allowed to overlap. By varying the diameter of the obstacles, we tune the average pore diameter, $\sigma_p$, of the medium, inspired by the experimental set-up of refs. [4,5]. The porous environment is characterized by the average pore diameter and the chord length. For ensemble averaging, we use 20 statistically independent structures.

We extract the average pore diameter of the medium by monitoring the transport behavior of Brownian particles with radius $\sigma_B$ in the porous environment. Therefore, we measure the mean-square displacement, $\langle|\Delta\mathbf{r}_B(t)|^2\rangle$ with $\Delta\mathbf{r}_B(t) = \mathbf{r}_B(t) - \mathbf{r}_B(0)$, and calculate the local exponent, $\alpha(t) = d\ln[\langle|\Delta\mathbf{r}_B(t)|^2\rangle]/d\ln t$, as a function of time. The local exponent displays a minimum at an intermediate time $\tau_{min}$, which allows introducing the pore diameter by $\sigma_p = \sigma_B + \langle|\Delta\mathbf{r}_B(\tau_{min})|^2\rangle^{1/2}$. For simplicity, we choose $\sigma_B = \sigma$.

To measure the chord length distribution $\varphi_{L_c}(\ell)$, we image several two dimensional planes randomly passing through the simulated porous medium (Fig. 1d). These images provide a map of the pore space. We then binarize these images where two phases represent solid obstacles and open pores, respectively. We calculate the distribution of chords of length $\ell$, which fit within each binarized pore space image. This protocol yields a direct measurement of straight pathways available in the pore space.

### Effect of pore shape.
We have addressed the effect of pore shape on the large-scale spreading of active agents. In particular, we have replaced the WCA potential [equation (9)], corresponding to pores with convex boundaries, and modeled the interaction between the polymer and concave pores by the interaction potential:

$$\frac{U_i^C}{k_BT} = -\epsilon_C e^{-(R_i/a)^{12}},$$

where $R_i$ is the distance between monomer $i$ of the polymer and the nearest obstacle. We choose $\epsilon_C = 100$ and $a = 3.7$, corresponding to concave voids of diameter $\sim 7.4\sigma$, as illustrated in Fig. 5c. We find that also for pores with concave walls the long-time behavior of the mean-square displacement is diffusive, see Fig. 5a. Extracting the long-time effective diffusivities shows that the non-monotonic behavior persists as a function of the reversal rate (Fig. 5b). The maximal diffusivity occurs at a reversal rate of $\lambda\tau_0 = 0.5$ and a run length of $\ell_{run} = 20\sigma$. This optimal behavior agrees with the findings for a porous

**a**      **b**      **c**

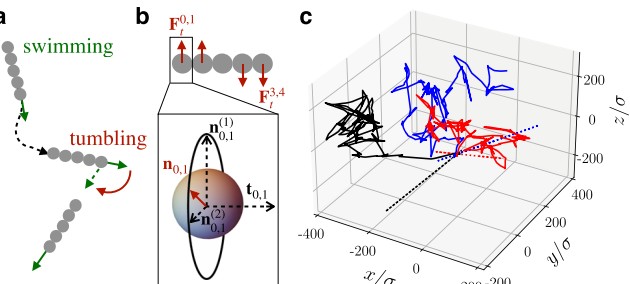

**Fig. 6 Run-and-tumble polymers in a dilute environment.** Sketch of the **a** run-and-tumble motion and **b** tumbling mechanism of a stiff polymer. Here, the unit vectors $\mathbf{n}_{0,1}^{(1)}$ and $\mathbf{n}_{0,1}^{(2)}$ span the plane normal to the tangent $\mathbf{t}_{0,1}$. The normal vector $\mathbf{n}_{0,1}$ lies in this plane. **c** Trajectories of three run-and-tumble polymers (solid lines) and run-reverse polymers (dotted lines) with vanishing translational diffusivity, $D_0 = 0$, and run-length $\ell_{run}/\sigma = 20$. Source data are provided as a Source Data file.

environments with convex pore shapes, as discussed in the main text. Overall, the effective diffusivities are smaller than for a convex environment, which may indicate that the polymer explores the individual pores for a longer time and should depend on the energy depth $\epsilon_C$.

### Effect of the re-orientation mechanism.
We further study the effect of the re-orientation mechanism of active polymers on their spreading in a porous environment. To complement our findings for run-reverse polymers, we consider a run-and-tumble polymer, which randomly changes its swimming direction at exponentially distributed tumbling events with tumbling rate $\lambda$ (Fig. 6a). At the tumbling event the active forces may change sign $\mathbf{F}_p \rightarrow \alpha\mathbf{F}_p$ with $\alpha$ randomly chosen from $\{-1, 1\}$ and the polymer is subject to tumbling torques. In particular, only at the tumbling event (hence during one time step) opposite forces $\mathbf{F}_t^{0,1}$ and $\mathbf{F}_t^{3,4}$ are applied to the zeroth and first and third and forth monomers, respectively (Fig. 6b), so that the polymer gets a random 'kick' leading to its re-orientation.

These 'random' tumbling forces are chosen (on average) perpendicular to the backbone of the polymer: $\mathbf{F}_t^{0,1} = F_t(\mathbf{n}_{0,1} + \mathbf{n}_{1,2})/2$ and $\mathbf{F}_t^{3,4} = -F_t(\mathbf{n}_{2,3} + \mathbf{n}_{3,4})/2$, where the force magnitude is $F_t = 2\cdot10^5 k_BT/\sigma \gg F_p$. As the polymer is stiff, we note that the forces approximately balance $\mathbf{F}_t^{0,1} + \mathbf{F}_t^{3,4} \simeq \mathbf{0}$.

To obtain the component of the force direction $\mathbf{n}_{0,1}$ (and similarly $\mathbf{n}_{1,2}$, $\mathbf{n}_{2,3}$, $\mathbf{n}_{3,4}$), we first define the plane normal to the tangent vector $\mathbf{t}_{0,1}$, which is spanned by the unit normal vectors $\mathbf{n}_{0,1}^{(1)}$ and $\mathbf{n}_{0,1}^{(2)} = \mathbf{t}_{0,1} \times \mathbf{n}_{0,1}^{(1)}$ (Fig. 6b). Then we choose $\mathbf{n}_{0,1}$ as a random direction in this plane via $\mathbf{n}_{0,1} = \cos(\beta)\mathbf{n}_{0,1}^{(1)} + \sin(\beta)\mathbf{n}_{0,1}^{(2)}$ with $\beta$ drawn from a uniform distribution $\mathcal{U}_{[0,2\pi]}$.

To illustrate our algorithm, we show typical trajectories for polymers with vanishing translational diffusivity, $D_0 = 0$, and run-length $\ell_{run}/\sigma = 20$, in Fig. 6c. The trajectories (solid lines) demonstrate that the polymer changes its swimming direction randomly at tumbling events and tumbles in 3D. We note that in this case reversing polymers only move back and forth along a straight line (dotted lines).

### Data analysis of the individual trajectories.
We extract the distributions for the trapping time, hopping time, and the hopping length from the individual trajectories. We note that the hopping time is defined as the time between hopping from one trap to the next or from one trapping event to the next reversal and the hopping length is the length the polymer moved during this time. To extract these quantities, we follow the approach from refs. [4,5] and first measure the average velocity of a non-tumbling polymer in a free environment, $\langle v\rangle$. Then we calculate the instantaneous velocities of the center monomer, $v_i = |\mathbf{r}_c(t_{i+1}) - \mathbf{r}_c(t_i)|/(t_{i+1} - t_i)$, where $t_i$ corresponds to the $i$-th time step, of individual particle trajectories. Thus, we can classify a hopping phase by $v_i \geq \langle v\rangle/3$ and a trapping phase by $v_i < \langle v\rangle/3$, which allows extracting the trapping and hopping time distributions, $\varphi_T(t)$ and $\varphi_H(t)$ with mean durations, $\tau_T$ and $\tau_H$. In addition, we keep track of the reversal times, which are input to our simulations, and compare it with the hopping duration. This provides the hopping length distribution, $\varphi_{\ell_H}(t)$ with mean hopping length $\langle\ell_H\rangle$. In particular, if the reversal time since the last trapping event $t_\lambda$ is shorter than the hopping phase $t_{hop}$, the hopping length corresponds to the length displaced until the reversal event. For $t_\lambda > t_{hop}$ the hopping length is the displacement from one trapping event to the next. For the comparison of the effective diffusivities extracted from simulations, $D_{eff}^{sim}$, to the predictions of the hop-and-trap model [equation (5)], we further use as effective velocity the cut-off velocity, $v = \langle v\rangle/3$. We can fully recover the non-monotonic behavior and explain the simulation data up to a constant pre-factor.

To test our approach, we have varied the cut-off velocity between $\langle v\rangle/5$ to $\langle v\rangle/2$ and found that it does not change our conclusions: the power-law behavior of the

trapping time distributions, the behavior of the hopping length distributions, and the semi-quantitative agreement of the effective diffusivities of the coarse-grained model and the simulations remain preserved.

**Entropic trap model for the trapping time distribution.** Motivated by other disordered media[4,5,41], the probability for the entropic trap $C$ is assumed to follow $P(C) = C_0^{-1} \exp(-C/C_0)$ with average trap depth $C_0$. By analogy to equilibrium physics, the probability of an active polymer to escape a trap of depth $C$ is assumed to obey an Arrhenius-like relation. Then the trapping duration is given by $t = \tau(\exp(C/X) - 1)$, where $X$ characterizes the active energy due to its swimming motion and $\tau$ corresponds to a characteristic time scale for trapping. In particular, the trapping duration vanishes, $t \to 0$, for small entropic traps, i.e., $C/X \to 0$. For a passive polymer the active energy $X$ is replaced by the thermal energy $k_B T$. The probability distribution of the trapping times can be obtained as $\varphi_T(t) = P(C)(\partial t/\partial C)^{-1} = \beta(1 + t/\tau)^{-1-\beta}/\tau$ with $\beta = X/C_0$.

**Renewal theory for the hop-and-trap dynamics.** The probability density for a particle to be in a hopping phase follows:

$$P_H(\Delta\mathbf{r}, t) = P_H^{(0)}(\Delta\mathbf{r}, t) + \int_{\mathbb{R}^3} d\boldsymbol{\ell} \int_0^t dt' H(\Delta\mathbf{r} - \boldsymbol{\ell}, t - t')\mathbb{P}_H(\boldsymbol{\ell}, t')\varphi_H^{(0)}(t'). \quad (11)$$

Here, $P_H^{(0)}(\Delta\mathbf{r}, t)$ denotes the probability that the particle has never been trapped before and the second term corresponds to the sum over all hopping phases, which started after at least one trapping event. Further, $\varphi_H^{(0)}(t) = \int_t^\infty dt' \varphi_H(t')$ is the probability that the hopping time exceeds $t$. The probability density (per time) that a new hopping phase starts obeys the equation of

$$H(\Delta\mathbf{r}, t) = H^{(1)}(\Delta\mathbf{r}, t) + \int_0^t dt' T(\Delta\mathbf{r}, t - t')\varphi_T(t'), \quad (12)$$

where $H^{(1)}(\Delta\mathbf{r}, t)$ is the probability that the particle starts the first hop. After a Fourier transform of the probability densities, $\Delta\mathbf{r} \to \mathbf{k}$, and by the convolution theorem, the renewal equations [Eqs. (3), (4), (11), (12)] simplify to

$$P_T(k, t) = P_T^{(0)}(k, t) + \int_0^t dt' T(k, t - t')\varphi_T^{(0)}(t'), \quad (13a)$$

$$P_H(k, t) = P_H^{(0)}(k, t) + \int_0^t dt' H(k, t - t')\mathbb{P}_H(k, t')\varphi_H^{(0)}(t'), \quad (13b)$$

$$T(k, t) = T^{(1)}(k, t) + \int_0^t dt' H(k, t - t')\mathbb{P}_H(k, t')\varphi_H(t'), \quad (13c)$$

$$H(k, t) = H^{(1)}(k, t) + \int_0^t dt' T(k, t - t')\varphi_T(t'). \quad (13d)$$

We further need to specify the probability densities

$$P_T^{(0)}(k, t) = (1 - p)\int_t^\infty dt' \varphi_T(t')(t' - t)/\tau_T, \quad (14a)$$

$$P_H^{(0)}(k, t) = p \, \mathbb{P}_H(k, t)\int_t^\infty dt' \varphi_H(t')(t' - t)/\tau_H, \quad (14b)$$

with $p = \tau_H/(\tau_H + \tau_T)$, which account for the fact that the system starts in a stationary state[42]. In particular, the probability to have never hopped before, $P_T^{(0)}(k, t)$, depends on the probability that the particle is in a trapped state, $1 - p$, and on the time integral, which represents the probability that the trapping phase exceeds time $t$. It can be rationalized as follows: The probability density that the trapping phase is of length $t'$ is given by $t'\varphi_T(t')/\tau_T$. The probability that after lag time $t$ the particle is still trapped is $(t' - t)\Theta(t' - t)/t'$, where $\Theta(\cdot)$ denotes the Heaviside function. Then the probability that the particle has not yet started a hopping phase at time $t$ is obtained by integrating over all durations: $\int_t^\infty dt' \varphi_T(t')(t' - t)/\tau_T$. Similarly, we can derive $P_H^{(0)}(k, t)$.

Moreover, the probability densities (per time) for the first trapping and hopping event are

$$T^{(1)}(k, t) = p \, \mathbb{P}_H(k, t)\int_t^\infty dt' \varphi_H(t')/\tau_H, \quad (15a)$$

$$H^{(1)}(k, t) = (1 - p)\int_t^\infty dt' \varphi_T(t')/\tau_T. \quad (15b)$$

Here, the probability for the first trapping event $T^{(1)}(k, t)$ depends on the probability that the particle is in a hopping state $p$ and has hopped for a time $t$ with propagator $\mathbb{P}_H(k, t)$. Further, the probability density for a hopping phase to be of length $t'$ is given by $t'\varphi_H(t')/\tau_H$. The probability for the lag time $t = 0$ to be in the same interval is uniformly distributed, $1/t'$. Then the probability density that the trapping phase starts at $t$ given a hopping phase of length $t'$ is obtained by the integral over all possible $t'$, $\int_t^\infty dt' \varphi_H(t')/\tau_H$. Similar considerations hold for $H^{(1)}(k, t)$.

Finally, we specify the propagator in the hopping phase $\mathbb{P}_H(k, t)$. Since we are interested in terms up to $\mathcal{O}(k^2)$, we expand it in $k$: $\mathbb{P}_H(k, t) \simeq 1 - k^2\langle|\Delta\mathbf{r}(t)|^2\rangle_{RR}/3!$, where the mean-square displacement of a run-reverse particle is $\langle|\Delta\mathbf{r}(t)|^2\rangle_{RR} = 2v^2/\tilde{\lambda}^2(\exp(-\tilde{\lambda}t) + \tilde{\lambda}t - 1)$. The effective rate depends on the turning angle $\vartheta_0$ via $\tilde{\lambda} = \lambda(1 - \cos\vartheta_0)$[9]. Subsequently, we perform a Laplace transform, $t \to s$, of equation (13a)–(13d) and use the convolution theorem to derive an analytical solution for the renewal equations in Fourier–Laplace space[42]. The formal solution has been presented elsewhere[42]. We insert the expansion of $\mathbb{P}_H$ and the hop and trapping time distributions, $\varphi_H$ and $\varphi_T$, into the theoretical predictions in Fourier–Laplace space and keep only terms up to $\mathcal{O}(k^4)$. Using the expansion from the main text, we derive an analytical solution for the mean-square displacement in Laplace space, $\langle|\Delta\mathbf{r}(s)|^2\rangle$. An analytical backtransform to time space is not possible, however, we can extract the long-time (corresponding to $s \to 0$) behavior, see main text.

We further note that the long-time effective diffusivity can also be derived analytically for a truncated power-law distribution, $\varphi_T(t) = \exp(-\gamma t)(1 + t/\tau)^{-1-\beta}f(\gamma\tau, \beta)$ with $f(\gamma\tau, \beta) = \left[e^{\gamma\tau}\beta\text{ExpIntE}(1 + \beta, \gamma\tau)\right]^{-1}$, where $\text{ExpIntE}(\cdot, \cdot)$ denotes the generalized exponential integral function[51]. It assumes the same form as equation (5) with average trapping time $\tau_T = \int_0^\infty t\varphi_T(t) \, dt = \tau\left[\text{ExpIntE}(\beta, \gamma\tau)/\text{ExpIntE}(1 + \beta, \gamma\tau) - 1\right]$. We note that the average trapping time of the truncated power-law distribution reduces for $\gamma = 0$ to that of the power-law distribution used in our manuscript. Details of the distribution become apparent in the short-time behavior of the mean-square displacement, but do not affect our data analysis for the long-time effective diffusivities.

## Data availability

Source data are provided with this paper in the Source Data file (Source_Data.zip). Source data are provided with this paper.

## Code availability

The computer code used for simulations is available from the corresponding authors upon reasonable request.

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

## Acknowledgements

Work by C.K. was funded by the Austrian Science fund (FWF) via the Erwin Schrödinger fellowship (Grant No. J4321-N27). The work was further supported by the NSF grant MCB-1853602 (H.A.S.). Work by S.M. and H.L. was supported by the Deutsche For-schungsgemeinschaft (Grant No. LO 418/23). Work by T.B. and S.S.D. was supported by NSF grant CBET-1941716, the Project X Innovation Fund, the Eric and Wendy Schmidt Transformative Technology Fund, a distinguished postdoctoral fellowship from the Andlinger Center for Energy and the Environment at Princeton University to T.B., and in part by funding from the Princeton Center for Complex Materials, a Materials Research Science and Engineering Center supported by NSF grants DMR-1420541 and DMR-2011750. For the purpose of open access, the authors have applied a CC BY public copyright licence to any Author Accepted Manuscript version arising from this submission.

## Author contributions

C.K., S.M., H.L., S.S.D., and H.A.S. designed the overall study. C.K. and S.M. performed the simulations, analyzed the data, and developed the coarse-grained theory. T.B. measured chord length distributions. S.S.D. and T.B. helped to design the analysis of hopping and trapping statistics. All authors discussed the results and implications. C.K. and S.M. wrote the manuscript with the help of all other authors.

## Competing interests

The authors declare no competing interests.
