## [Peer Review File · Nature Communications]

Reply to referees

A Geometric Criterion for the Optimal Spreading of Active Polymers in Porous Media

Christina Kurzthaler, Suvendu Mandal, Tapomoy Bhattacharjee, Hartmut Löwen, Sujit S. Datta, Howard A. Stone
Nat. Commun., NCOMMS-21-16606

1. Reply to Referee #1

Referee #1: The authors perform three-dimensional simulations of active stiff polymers in a background of overlapping sphere obstacles to represent the bacteria recently studied moving through porous media. When the dynamics of the active particles include direction reversals at some rate, they find that the translational diffusivity is strongly enhanced. They show that there is a geometric condition giving the greatest transport when the run length of the active polymer matches the longest straight path in the medium. To prove that it is the path length and not the pore shape which is important, they show that the same behavior appears when the pores are concave instead of convex. They connect the response of the system to an entropic trapping effect.

This study is well motivated, clearly presented, and makes an important contribution to the field. The authors take into account both the fully three-dimensional motion and the shape of the particles, which makes the simulations both more challenging and closer to the experimental conditions. The measures are clear and convincing, and the identification of a geometric criterion for optimal transport is particularly important. I recommend the paper for publication in its present form.

Reply: We are pleased by the very positive evaluation of our manuscript and that the referee values our work as '*well motivated, clearly presented*' and as an '*important contribution to the field*' warranting publication in its present form in Nature Communications.

2. Reply to Referee #2

Referee #2: This work studies how stiff polymers spread in 3D porous media numerically. The polymers move with a run-reverse mechanism and are introduced as a model for the motion of some motile bacteria. The authors find that the polymers spread non-monotonically with the reversal rate, enhancing the long-term diffusivity of the polymers in the porous structure under optimal reversal rates. A theory is presented to reproduce the results observed numerically and explain them in terms of a geometric criterion based on characteristic lengths of the pores and of the polymer's runs.

The manuscript is well written and clear. The work is of interest and convincing and I think deserves publication in Nature Communications after a few minor points are addressed:

Reply: We would like to thank the Referee for the very pleasing evaluation and for valuing that our manuscript 'deserves publication in Nature Communications'. We further thank the Referee for the constructive suggestions, which we gladly address below.

Referee #2: - non-monotonic behaviours are a common characteristic of active matter moving in porous environments or through obstacles. The authors should acknowledge and stress this more prominently in the introduction (most of the relevant literature is already referenced by the authors in the manuscript anyway).

Reply: We gladly address the point raised by the referee and added the following paragraph to the introduction:

'A non-monotonic transport behavior as a function of the tumbling rate has been found also theoretically in 2D systems [32,35,36], where the effects of pore shape [36] and mobile obstacles [32] have been addressed.'

Referee #2: - in the authors's model, polymers can only reverse in the direction they came from. Many bacteria tumble in 3D. How would this different mechanism (that also changes the 3D orientation) affect the authors' results?

Reply: We gladly respond to this remark, which was also emphasized by the editor. To address this point, we have performed additional simulations of run-and-tumble polymers, which change their swimming direction randomly in 3D at the tumbling event (see Fig. 1a-b for a sketch).

Fig. 1c shows typical trajectories of run-and-tumble polymers with vanishing translational diffusivity, $D_0 = 0$, and run-length $\ell_{\text{run}}/\sigma_0 = 20$. The trajectories demonstrate that the polymer changes its swimming direction randomly at tumbling events and tumbles in 3D. We note that in this case reversing polymers only move back and forth along a straight line.

In dense porous environments, we find, as for run-reverse polymers, that the run-and-tumble polymers exhibit hop-and-trap dynamics (see Fig. 2a) and that the long-time effective diffusivities display a non-monotonic behavior as a function of the tumbling rate. The corresponding mean-square displacements for $N = 1000$ are shown in Fig. 2b. Most importantly, the optimal behavior obeys our geometric criterion (see Fig. 2c).

We have included our findings for run-and-tumble polymers in our revised manuscript in the following way:

- In the appendix, we now describe the method for simulating run-and-tumble polymers and included Fig. 1 (corresponding to Fig. 6 in the manuscript):

Figure 1: **Run-and-tumble polymers in a dilute environment.** Sketch of the **a** run-and-tumble motion and **b** tumbling mechanism of a stiff polymer. Here, the unit vectors $\mathbf{n}_{0,1}^{(1)}$ and $\mathbf{n}_{0,1}^{(2)}$ span the plane normal to the tangent $\mathbf{t}_{0,1}$. The normal vector $\mathbf{n}_{0,1}$ lies in this plane. **c** Trajectories of three run-and-tumble polymers (solid lines) and run-reverse polymers (dotted lines) with vanishing translational diffusivity, $D_0 = 0$, and run-length $\ell_{\text{run}}/\sigma_0 = 20$.

Figure 2: **Run-and-tumble polymers in porous environments.** **a** Representative (1D) displacements, $\Delta x_c(t)$, of polymers with different reversal rates, λ , as a function of time. Horizontal solid lines indicate the trapping phases for different λ . (*Inset*) Particle trajectories of the center monomer of a rarely reversing polymer, $\lambda\tau_0 = 5 \cdot 10^{-4}$ (purple), and moderately reversing polymers, $\lambda\tau_0 = 0.5$ and 15 (orange and green, respectively). The trajectories are shown in the xy plane, $(\Delta x_c(t), \Delta y_c(t))$. **b** Mean-square displacements of run-and-tumble polymers in a porous environment with $N = 1000$ obstacles. The Péclet number is $\text{Pe} = 50$. **c** Rescaled effective diffusivities, $D_{\text{eff}}^{\text{sim}}$, of run-reverse (RR) and run-and-tumble (RT) polymers for $\text{Pe} = 50$ and different N . The data are rescaled by the maximal pore length $L_{c,\text{max}}$ extracted from the chord-length distributions in Fig. 1e of our manuscript.

‘We further study the effect of the re-orientation mechanism of active polymers on their spreading in a porous environment. To complement our findings for run-reverse polymers, we consider a run-and-tumble polymer, which randomly changes its swimming direction at exponentially distributed tumbling events with tumbling rate λ [Fig. 6a]. At the tumbling event the active forces may change sign $\mathbf{F}_p \rightarrow \alpha \mathbf{F}_p$ with α randomly chosen from $\{-1, 1\}$ and the polymer is subject to tumbling torques. In particular, only at the tumbling event (hence during one time step) opposite forces $\mathbf{F}_t^{0,1}$ and $\mathbf{F}_t^{3,4}$ are applied to the first and second and third and fourth monomers, respectively [Fig. 6b], so that the polymer gets a random ‘kick’ leading to its re-orientation.

These ‘random’ tumbling forces are chosen (on average) perpendicular to the backbone of the polymer: $\mathbf{F}_t^{0,1} = F_t (\mathbf{n}_{0,1} + \mathbf{n}_{1,2}) / 2$ and $\mathbf{F}_t^{3,4} = -F_t (\mathbf{n}_{2,3} + \mathbf{n}_{3,4}) / 2$, where the force magnitude is $F_t = 2 \cdot 10^5 k_B T / \sigma \gg F_p$.

To obtain the component of the force direction $\mathbf{n}_{0,1}$ (and similarly $\mathbf{n}_{1,2}$, $\mathbf{n}_{2,3}$, $\mathbf{n}_{3,4}$), we first define the plane normal to the tangent vector $\mathbf{t}_{0,1}$, which is spanned by the unit normal vectors $\mathbf{n}_{0,1}^{(1)}$ and $\mathbf{n}_{0,1}^{(2)} = \mathbf{t}_{0,1} \times \mathbf{n}_{0,1}^{(1)}$ [Fig. 6b]. Then we choose $\mathbf{n}_{0,1}$ as a random direction in this plane via $\mathbf{n}_{0,1} = \cos(\beta) \mathbf{n}_{0,1}^{(1)} + \sin(\beta) \mathbf{n}_{0,1}^{(2)}$ with β drawn from a uniform distribution $\mathcal{U}_{[0,2\pi]}$.

To illustrate our algorithm, we show typical trajectories for polymers with vanishing translational diffusivity, $D_0 = 0$, and run-length $\ell_{run} / \sigma_0 = 20$, in Fig. 6c. The trajectories (solid lines) demonstrate that the polymer changes its swimming direction randomly at tumbling events and tumbles in 3D. We note that in this case reversing polymers only move back and forth along a straight line (dotted lines).’

- We have added Fig. 2c to our main text (corresponding to Fig.3c) and included the following paragraph in the main text:

‘Furthermore, we have addressed the effect of run-and-tumble motion on the overall spreading of active polymers, where, instead of reversing, the swimming direction after the tumbling event is random. Our results [Fig. 3c] demonstrate that our geometric criterion remains valid. We observe that the overall diffusivities increase for $N = 1000$ with respect to those of run-reverse polymers, as 3D tumbling allows polymers to spread further. Most importantly, in dense porous environments ($N = 1200$) the effective diffusivities of run-and-tumble polymers collapse with those of run-reverse polymers. This indicates that the overall spreading in dense porous environments is characterized by hop-and-trap dynamics, which are fully determined by the geometry, irrespective of the re-orientation mechanism.’

- We have also added a note to our conclusions:

‘In particular, we demonstrate that this criterion remains valid in dense environments for different reorientation mechanism of the active polymers.’

- We have also emphasized our findings for run-and-tumble polymers in the abstract

‘More significantly, our criterion unifies results for porous media with disparate pore sizes and shapes and for run-and-tumble polymers.’

and the introduction

‘... this large-scale non-monotonic behavior persists irrespective of the pore shapes and reorientation mechanism of the polymers...’

Referee #2: - the authors mention that their results show a non-monotonic spreading behaviour also in concave porous environments differently from previous work. I don’t think this is a fair comparison as this will ultimately depend on the density of pores and indeed in Fig. 2b the authors show that at low

density the non-monotonic trend fades away. The authors should stress the importance of pore density also when comparing with previous literature.

Reply: We are grateful for the valuable comment. To clarify this point, we have added the following sentence to our conclusions:

'The non-monotonic behavior fades for low packing fraction, which corroborates earlier predictions for dilute porous environments with concave pore shapes [36].'

Referee #2: - Fig. 4b shows truncated power laws rather than power laws. The authors added a note in the methods on how their theory and results would change if truncated power laws are considered rather than power laws. It would be useful to discuss this directly in the main text.

Reply: We thank the referee for this remark and now discuss the truncated power law distribution also in the main text:

'We note that for a power-law distribution truncated at rate γ , $\varphi_T(t) \sim \exp(-\gamma t)(1 + t/\gamma)^{-1-\beta}$, which may be more appropriate to describe our data [Fig. 4b], the effective diffusivity assumes the same form as in Eq. (5) with average trapping time $\tau_T \equiv \tau_T(\gamma, \tau, \beta)$ depending on the parameters γ , τ , and β (see Methods).'